# Resistance training leading to repetition failure increases muscle strength and size, but not power-generation capacity in judo athletes

**Miyuki Nakatani**○*, **Yohei Takai**○, **Hiroaki Kanehisa**

National Institute of Fitness and Sports in Kanoya, Kanoya, Kagoshima, Japan

* nakatani@nifs-k.ac.jp

**Data Availability Statement:** All relevant data are within the manuscript and its Supporting information files.

## Abstract

Strength-trained athletes has less trainability in muscle size and function, because of their adaptation to long-term advanced training. This study examined whether resistance training (RT) leading to repetition failure can be effective modality to overcome this subject. Twenty-three male judo athletes completed a 6-week unilateral dumbbell curl training with two sessions per week, being added to in-season training of judo. The participants were assigned to one of three different training programs: ballistic light-load (30% of one repetition maximum (1RM)) RT to repetition failure ($RF_{LB}$) (n = 6), traditional heavy-load (80% of 1RM) RT to repetition failure ($RF_{HT}$) (n = 7), and ballistic light-load (30% of 1RM) RT to non-repetition failure ($NRF_{LB}$) (n = 10). Before and after the intervention period, the muscle thickness (MT) and the maximal voluntary isometric force (MVC) and rate of force development ($RFD_{max}$) of elbow flexors were determined. In addition, theoretical maximum force ($F_0$), velocity ($V_0$), power ($P_{max}$), and slope were calculated from force-velocity relation during explosive elbow flexion against six different loads. For statistical analysis, p < 0.05 was considered significant. The MT and MVC had significant effect of time with greater magnitude of the gains in $RF_{HT}$ and $NRF_{LB}$ compared to $RF_{LB}$. On the other hand, all parameters derived from force-velocity relation and $RFD_{max}$ did not show significant effects of time. The present study indicates that ballistic light-load and traditional heavy-load resistance training programs, leading to non-repetition failure and repetition failure, respectively, can be modalities for improving muscle size and isometric strength in judo athletes, but these do not improve power generation capacity.

## Introduction

Ability to generate high muscular power is essential to success in many athletic and sporting activities [1, 2]. In general, resistance training (RT) consisting of ballistic or explosive exercises are considered highly specific to maximal power movements and develops many of the components of the neuromuscular systems facilitating such actions [3]. There are two main schools

**Funding:** The author(s) received no specific funding for this work.

**Competing interests:** The authors have declared that no competing interests exist.

of thought concerning the optimum loads for power training [1]: one is the use of light-load (<50% of one repetition maximum [1RM]) and the other heavy-load (50–70% of 1RM). Some earlier studies have demonstrated that training loads of <50% of 1RM or force level attained by maximal voluntary contraction (MVC) may be effective to optimize power-generating capacity over a wide load range [4–6]. Moss et al. [6], who examined effect of elbow flexion RT with maximal effort at each of three different loads (15%, 35%, and 90% of 1RM) on muscle size and function of elbow flexors in well-trained men, have shown that the RT at 35% of 1RM produced a similar significant increase in power at all loads ranging 15% to 90% of 1RM, whereas RT at 90% of 1RM showed load specificity in the effect of the training. Similarly, McBride et al. [5] also have reported that for strength-trained athletes, the jump squat training with 30% of 1RM significantly increased peak power in jump squats with loads of 30%, 55%, and 80% of 1RM. In their results, the training at 80% of 1RM did not produce a significant gain in power output at 30% of 1RM. These findings suggest that in both single- and multi-joints RT protocols, the use of light-load (30–35% of 1RM), yielding near maximal power output in load-power relationship, would increase power efficiently over a wide load range [6]. However, the level of load at which maximal power output appears in load-power relationship varies among athletic events [7]. It is unknown whether the previous findings cited here can be applied to athletes with greater muscularity and strength and power generation capacity such as judo athletes [8].

As compared to the untrained individuals, athletes have less trainability in muscle size and function, because of the advanced training stimulation over long-term in them [9]. A training modality, which could potentially maximize muscle size and function and consequently power generation capacity in athletes, is RT leading to "repetition failure (or muscular failure) [10]", i.e., the inability to perform another concentric repetition while maintain proper form [11]. A physiological background for this idea is that heavy-load RT leading to repetition failure can activate the motor units of the exercising muscles more than heavy-load RT under non-repetition failure condition or light-load RT leading to repetition failure [12–15]. Considering the contribution of neural adaptation to strength gains [16–18], it is assumed that as similar as ballistic light-load RT, traditional heavy-load RT leading to repetition failure would also be a potential modality to maximize power generation capacity in strength-trained athletes by resulting in neuromuscular adaptation contributing to develop greater muscular strength. A few studies have examined the effect of resistance training leading to repetition failure on power generating capacity in athletes [19, 20].

In untrained individuals, when the exercises are performed until the point of repetition failure and the training volume (loads × number of repetitions per set × number of sets) are equated among the protocols, the effects of RTs on muscle size are independent of the training load ranging 30% to 80% of 1RM [11, 21]. In recreationally trained individuals, too, muscle hypertrophic gains are similar regardless of low- (>15 RM), moderate- (9–15 RM), and high-load (≦8 RM) RTs when exercises are performed to the point of repetition failure [22]. On the other hand, strength gain is greater in heavy-load than in light-load RT [11, 22, 23]. For athletes, however, previous findings on the differences between the effects of repetition failure and non-failure protocols are very scarce and these are still in controversial [19, 20]. Drinkwater et al. [19] observed that RT-induced gains in strength and power for elite junior athletes were greater in repetition failure protocol than in non-repetition failure protocol. From the findings of Izquierdo et al. [20], gains in 1RM and power of 11-wk RT training period in Basque ball players were similar between the repetition failure and non-repetition protocols. In their results, however, an identical 5-wk peaking period of maximal strength and power training, being set after the completion of the 11-week training period, produced larger gains in muscle power output in non-repetition failure group compared to repetition failure group.

The reasons for the discrepancy between the two studies cited above are unknown but might involve the differences in the physical characteristics of the participants, the training programs adopted and/or the event-related competitive and training activities. In any case, available information on the effects of RT leading to repetition failure on muscle size and function in athletes is quite limited. The present study aimed to elucidate these subjects. To this end, we set three different RT programs for elbow flexors of judo athletes, consisting of unilateral dumbbell curl: ballistic light-load (30% of 1RM) RT leading to repetition failure ($RF_{LB}$), traditional heavy-load (80% of 1RM) RT leading to repetition failure ($RF_{HT}$), and ballistic light-load (30% of 1RM) RT with non-repetition failure ($NRF_{LB}$) by adopting a cluster set configuration [24]. We treated $NRF_{LB}$ as a control group and examined the influences of repetition failure on muscle size and function through the comparisons between $NRF_{LB}$ and either $RF_{LB}$ or $RF_{HT}$. We hypothesized that 1) gains in muscle size in $RF_{LB}$ and $RF_{HT}$ are similar between the two programs and greater than that in $NRF_{LB}$, 2) gain in muscle strength is greater in $RF_{HT}$ than in the other programs, and 3) gain in maximal power is greater in $RF_{LB}$ and $RF_{HT}$ than $NRF_{LB}$.

## Materials and methods

### Experimental approaches to the problem

Judo athletes were adopted as the participants to examine whether the RT leading to repetition failure would be effective on muscle size, strength and joint power in athletes, having greater musculature and muscle strength and power generation capacity [8, 25]. Three training programs were designed to combine load (30 or 80% of 1RM) and fatigue condition (repetition failure or not repetition failure). The detail of the training program is described later. In the two light-load programs, the participants were asked to conduct concentric phase with ballistic contraction (maximum voluntary contraction velocity) mode. On the other hand, the participants involved in the high-load program performed the exercise task at a constant tempo. We designed RT program leading to repetition failure against load of 30% or 80% of 1RM and that leading to non-repetition failure against load of 30% of 1RM.

### Subjects

Twenty-six male judo athletes voluntarily participated in this study (Fig 1). All participants recruited in 1st March to 31st Aug 2020. The means and standard deviations (SDs) for age, body height, and body mass of the participants were 20.0 ± 0.7 years, 170.4 ± 5.3 cm, and 78.0 ± 10.1 kg, respectively. All participants had experienced judo-specific competitive and training activities for 10 years or more. Judo training and resistance training were performed 5–6 and 2–3 days per week, respectively. During the experimental period, resistance training targeting the elbow flexors was not performed. Based on the classification reported in Mackay et al., their athlete taxonomy corresponded to Tier 3 or Tier 4 [26]. They had participated in intercollegiate or international judo competitive tournaments in the preceding year. The participants were randomly assigned to one of three training programs: $RF_{LB}$ (n = 8), $RF_{HT}$ (n = 8), and $NRF_{LB}$ (n = 10). Three individuals were excluded from the analysis of this study (missed two training sessions (n = 1), weight loss for participation in judo match (n = 1), and had outlier data (n = 1)). Therefore, the analyzed data were those obtained from the 23 participants ($RF_{LB}$, n = 6; $RF_{HT}$, n = 7; $NRF_{LB}$, n = 10). The ethical committee of the local university approved this study (the National Institute of Fitness and Sports in Kanoya's Ethics Committee #11–102). Prior to the start of the present study, all participants were informed of the purpose, the risks and benefits of this study and procedures used in this study. All participants gave their written informed consent for participation in the study.

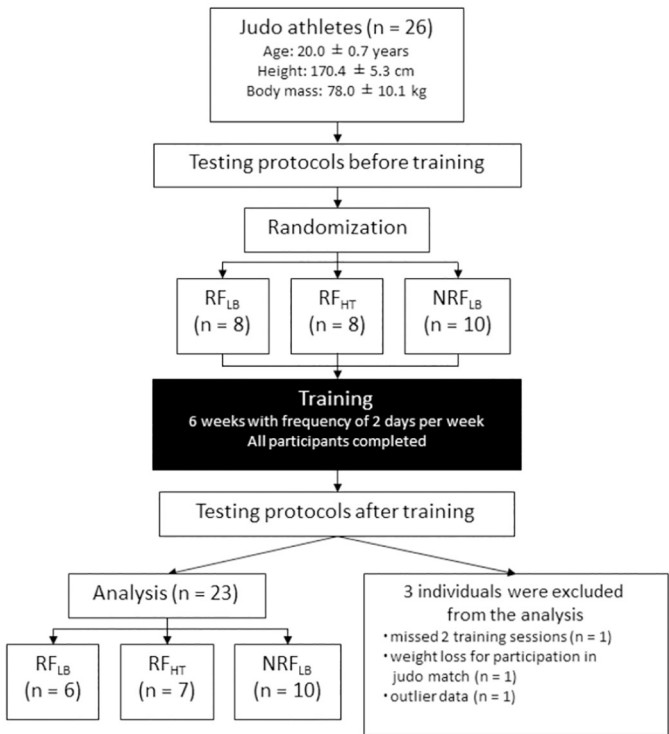

**Fig 1. Flow diagram and demographic information of participants.**

## Procedures

**Training programs.** All training groups completed a 6-weeks of unilateral dumbbell curl training. The prescribed program was conducted during the in-season phase of judo. During the intervention period, all participants were on judo-specific training program, being conducted five days per week more than 2 hours per day. The judo-specific training did not include RT programs. All participants performed the unilateral dumbbell curl training as an adjunct exercise task to judo-specific training for 6 weeks with frequency of 2 days per week. They selected the right or the left arms as their training limbs, in relation to their own patterns of "kumite" which is technique of gasping opponent's jacket in judo-competitive and -specific training activities. In the execution of dumbbell exercise, the participants were instructed to perform the task in standing position with their elbows and posterior parts of upper arms on a wall to prevent leaning upper body either forward or backward. They lifted the dumbbell from approximately 40 deg flexed position (fully extension = 0 deg) to fully flexed elbow joint. The content for each of the three training programs is as follows:

$RF_{LB}$. Participants performed a unilateral dumbbell curl with load of 30% 1RM to repetition failure. They were instructed to perform the concentric phase (lifting the load) as fast and forceful as possible and the eccentric phase (lowering the load) for 4 seconds with the aid of an audible metronome. In addition, the participants were asked to repeat the prescribed exercise three times with an examiner's assistance after reaching to the condition of the inability to perform another concentric repetition.

$RF_{HT}$. Participants performed a unilateral dumbbell curl with load of 80% 1RM to repetition failure. They were instructed to perform each of the concentric and eccentric phases for 2 seconds with the aid of an audible metronome. In addition, the participants were asked to

repeat the prescribed exercise three times with an examiner's assistance after reaching to the condition of the inability to perform another concentric repetition.

*NRF$_{LB}$.* Participants performed a unilateral dumbbell curl 20 repetitions per set with a 30 s rest between each 5 repetitions. The training intensity was 30% of 1RM. The participants were instructed to perform the concentric phase as fast and forceful as possible ($<$ 1 second) and the eccentric phase for 4 seconds with the aid of an audible metronome.

Each of the training sessions was initiated with a warm-up consisting of 10 repetitions with a dumbbell shaft (2 kg). All groups conducted three sets per session with a 2-min rest interval between sets. The training load was adjusted every 2 weeks based on the results of 1RM measurement of single arm dumbbell curl. A strength and conditioning researcher supervised each workout session carefully and recorded the compliance and individual workout data during each training session so that exercise prescriptions were properly administered during each training session (e.g., number of repetitions, rest, and tempo of movement). In the present study, the training volume was defined as the product of absolute load (kg), total number of repetitions (rep), and total time required for concentric and eccentric phases (s). A metronome was used to describe the tempo during the exercises to 60 bpm. When participants could no longer maintain the tempo within 1 bpm, the training for each set was discontinued. In NRF$_{LB}$ and RF$_{LB}$ with ballistic contraction, the contraction duration was less than 1 second so that lifting time was considered to be 1 second.

**Testing protocols.**   Participants completed testing protocols before initiating training (pre-training) and after the completion of 6 weeks of training (post-training). They had 3–5 days of recovery between their last training session and the post testing session. Participants were thoroughly familiarized to all test procedures before the actual assessment. The testing protocols involved the measurements of muscle thickness (MT) of elbow flexors, 1RM of single arm dumbbell curl, MVC and rate of force development (RFD) during isometric elbow flexion, and force-velocity relationship (F-V relation) of elbow flexors. Experimental setup and procedures of the test protocols were the same as those adopted in a prior study [8].

*Measurements of MT.* After the completion of anthropometric measurements, MT at the anterior part of the upper arm was determined by using a brightness-mode ultrasound apparatus (ProSound Alpha6, Hitachi Aloka Medical, Japan) with a linear-array probe (7.27 MHz). The procedures for obtaining ultrasonographic image and for determining MT from the image were identical to those described in an earlier study [27]. Briefly, the MT measurement was conducted at 60% of the upper arm length defined as the distance from the acromial process to the lateral epicondyle of the humerus. During the MT measurement, the participants stood upright with their arms relaxed and extended. The probe was placed perpendicular to the skin without depressing the dermal surface and a probe was coated with water-soluble transmission gel, which provided acoustic contact. The MT was defined as the distance from the subcutaneous adipose tissue-muscle interface to the muscle-bone interface. The muscles involved in the measured MT were the biceps brachii and brachioradialis. All images were analyzed by using image analysis software (Image J ver. 1.47, NIH, USA). We calculated muscle cross-sectional area index (CSA$_{index}$) of the elbow flexors by using the following equation [27]:

$$CSA_{index} = \pi \times (MT/2)^2$$

where $\pi$ is a constant, 3.14159, and MT is in cm.

*Measurements of 1RM.* After the completion of the standardized warm up, the participants lifted a dumbbell with a given weight from fully extended with same posture during training. The participants were instructed to perform a purely concentric action. The load lifted in the

best successful attempt was recorded in kilograms and adopted as 1RM. The load was adjusted by 0.5 kg.

*Measurements of MVC and RFD.* The isometric and dynamic tasks were performed using a custom-made dynamometer with tension/compression load cells (TR22S, SOHGOH KEISO CO., LTD, Japan). Participants were seated on an adjustable chair with the shoulder, and hip joints flexed at 90˚. Their hips and shoulders were fixed to backrests of chairs, and wrists were fixed to lever arms of the dynamometer in a neutral position by non-elastic belts. The rotation axis of the elbow joint was visually aligned as closely as possible with that of the dynamometer. The forearm was fixed to the lever arm that could rotate freely around the axis with the wrist joint kept in a neutral position. The force signals during the tasks were amplified and attenuated with a low-pass filter (<100 Hz, DPM-912B, KYOWA, Japan). The axis of the potentiometer's lever arm was equipped with a dynamometer to detect voltage changes associated with those in the elbow joint angles during the dynamic contraction task. The voltage signals were converted to angle (deg) from the voltage-angle relationship. The force and angle signals were sampled at a frequency of 2kHz via a 16-bit analog/digital converter (PowerLab/16s: AD Instruments Sydney, Australia) and stored on a personal computer. All measurements were conducted by the same investigator (MN).

*MVC.* In MVC task, the elbow joint was held at a 40˚ flexed position (0˚ corresponds to full elbow extension). The participants were instructed to exert gradually elbow flexion force from the baseline to the maximum level and sustain at the maximum for approximately 2 s. After a standardized warm-up protocol (50% and 80% of subjective maximal effort) and familiarization with the measurement apparatus, two MVC trials were performed with a 3-min interval between the trials. If the difference between the two trials in the measured forces was more than 10%, the MVC trial was made again. The highest value among the 2 or 3 isometric forces was adopted as MVC force and used to determine the load set in the dynamic contraction task.

*RFD.* The participants performed a series of 10 explosive isometric contractions with an interval of 15 s between the contractions at a 40˚ flexed position of the elbow joint. Participants were instructed to flex their elbows 'as fast as and as hard as possible' with an emphasis on 'fast' from a relaxed state. Contractions involving a visible countermovement or pre-tension were discarded and another attempt was made. During each explosive contraction, participants were asked to exceed 80% of MVC, which was depicted by an on-screen marker. Onset of muscle contraction was defined as the time point at which the developed force curve exceeded 2.5% of MVC. The peak RFD was calculated as the maximum value in the first derivative of force overtime on the filtered signals during the contraction [28] and referred to as $RFD_{max}$. The top three among the calculated $RFD_{max}$ values were averaged and adopted as the representative value of the $RFD_{max}$.

*Measurements of F-V relation.* After a 5-min rest following the completion of the MVC and RFD tasks, the participants were asked to perform the dynamic contraction task consisting of ballistic contractions against three different loads in a random order (unload condition, 30, and 75% of MVC). They were asked to flex the elbow joint as strongly and quickly as possible in each of the three load conditions. The participants' position and the fixation of the body during the dynamic contraction task were identical to those during the MVC tasks. Weights were attached to pulley moving in conjunction with the lever arm, and the range of the motion was from 40˚ to 120˚ of the elbow joint angle. A shock absorber was put on the portion at 120˚. Prior to the execution of each trial, an examiner lifted the lever arm until the start position (corresponded to 40˚) on checking raw data of joint angle with a monitor visually. At the starting position, the participants were kept to relaxed condition by supporting the load by the examiner until the start of elbow flexion with maximal effort. Participants were informed that

the magnitude of the load had been set in advance. Rest intervals of at least 30 s and 3 min respectively were set between trials in a given load condition and between loads sets. The analysis of elbow flexion force and velocity at each load condition is described in detail below.

*Determinations of force (F) and velocity (V)*. By analyzing the data obtained through the dynamic contraction task, we calculated velocity, force, and power output during ballistic elbow flexions against the three different loads. First, we obtained angular velocity by differentiating the angle by time, and then converted it to the tangential velocity (the elbow flexion velocity, m/s) by multiplying the perpendicular distance between the load cell and the lever-arm axis of the dynamometer. Secondly, we averaged force and velocity variables over a range of elbow joint angles from 40˚ to 100˚ and used as functional variables developed for the specific load condition. We referred to the force and velocity as F and V, respectively.

*Calculation of theoretical maximum force ($F_0$), velocity ($V_0$), power ($P_{max}$), and slope*. We calculated the $F_0$, $V_0$, and $P_{max}$ as representative variables indicating F-V relation across the three different loads. We defined the points of intersection of the regression line with the ordinate and transversal axis as $F_0$, and $V_0$, respectively, and calculated $P_{max}$ as described in an earlier study [29, 30] by using the following equation:

$$P_{max} = F_0 \times V_0/4$$

In addition to the absolute values, we expressed $F_0$ and $P_{max}$ as values relative to $CSA_{index}$ ($F_0/CSA_{index}$ and $P_{max}/CSA_{index}$). Furthermore, we adopted the slope of the regression line for the F-V relation (F-$V_{slope}$) as a parameter indicative of predominance of force (or velocity) in the relationship [30].

## Statistics

We have presented descriptive data as means ± SDs. We used a one-way analysis of variance (ANOVA) to examine group differences in the measured variables at baseline and to test the main effect of group on total training volume and total number of repetitions. When the main effect of group was significant, we used a Bonferroni test to examine the difference between groups in the corresponding variables. Moreover, a two-way repeated measures ANOVA (2 times × 3 groups) was used to test the main effects of time and its interaction with group on the measured variables. At baseline, there were no significant differences in the measured variables among the three training groups. Sphericity was checked by Mauchly's test in ANOVA, and *P* values were modified with Greenhouse-Geisser correction when necessary. We calculated Hedges' g [31] with 95% confidence intervals as indices of effect sizes. We interpreted the index as large: ≥0.80, medium: 0.50–0.79, small: 0.20–0.49, or trivial: <0.20 [31]. We also calculated Pearson's product-moment correlation coefficient (*r*) to examine the associations between F and V. We set the level of significance as $p < 0.05$. We analyzed all the data using SPSS software (SPSS statistics 27; IBM, Japan).

## Results

### Total number of repetitions and training volume

There were significant effects of group in the total values over the intervention period in the number of repetitions (Fig 2A, F = 214.6, p<0.001), and training volume (Fig 2B, F = 70.1, p<0.001). The two variables were significantly greater in $RF_{LB}$ than in $RF_{HT}$ (p<0.001) and $NRF_{LB}$ (p<0.001) and in $NRF_{LB}$ than in $RF_{HT}$ (p<0.001).

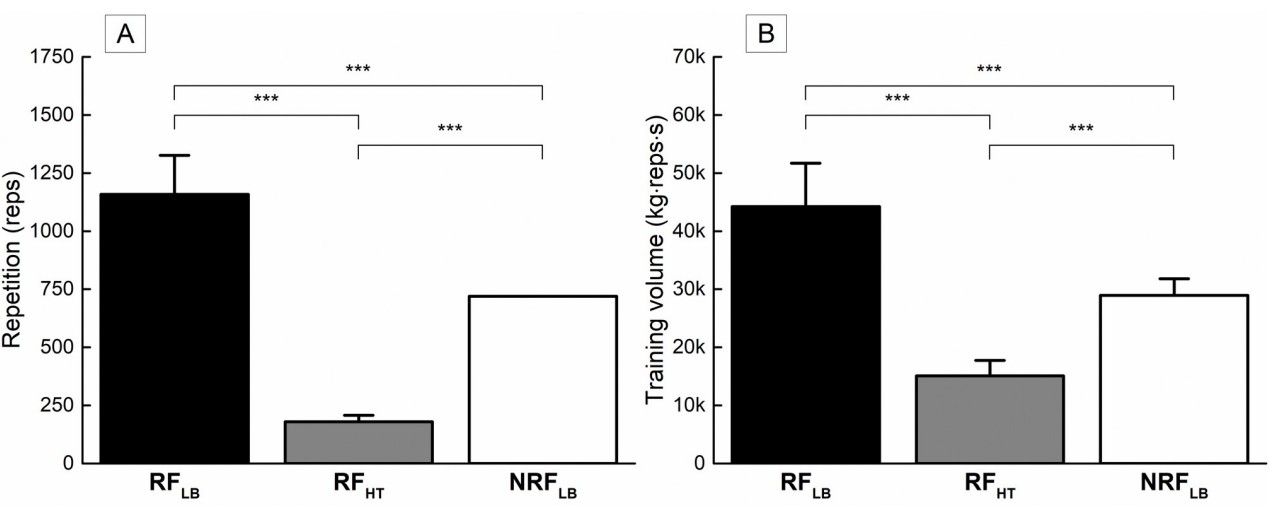

**Fig 2.** Comparisons on total number of repetitions (A) and total training volume (B) over intervention period. *** indicates the difference between the groups is significant at p < 0.001.

### 1RM, MT, and CSA$_{index}$

Fig 3 shows changes in 1RM (Fig 3A) and MT (Fig 3B). Descriptive data on 1RM, MT and CSA$_{index}$ are summarized in Table 1. 1RM had significant effect of time without significant interaction of time and group. The magnitude of effect size (Hedges' g) for the change in 1RM was large in all groups. MT had a significant effect of time without significant interaction of time and group. The magnitude of effect size for change in MT was large in RF$_{HT}$ and NRF$_{LB}$, but small in RF$_{LB}$.

### MVC, MVC/CSA$_{index}$, and RFD$_{max}$

MVC (Fig 4A) had a significant effect of time without significant interaction of time and group (Table 2). The magnitude of effect size in MVC was medium in RF$_{HT}$ and NRF$_{LB}$ and small in RF$_{LB}$. On the other hand, MVC/CSA$_{index}$ (Fig 4B) had no significant effect of time and interaction of time and group. Similarly, RFD$_{max}$ also did not have significant effect of time and interaction of time and group (Table 2).

### Parameters of F-V relation

Descriptive data on the parameters derived from F-V relation are summarized in Table 3. Each of F$_0$, V$_0$, P$_{max}$, and F-V$_{slope}$ had no significant effect time and interaction of time and group.

F$_0$/CSA$_{index}$ and P$_{max}$/CSA$_{index}$ had significant effects of time without significant interactions of time and group (Table 3, Fig 5). The magnitude of effect size for the negative changes was large in RF$_{HT}$, medium in NRF$_{LB}$, and small in RF$_{LB}$ (Table 3).

## Discussion

While the three different training programs adopted here increased the size and force generation capacity of elbow flexors in judo athletes, these did not improve power generation capacity, regardless of the condition of muscle failure and magnitude of loads. The current results deny the hypothesis set at the start of this study. However, it should be noted that the

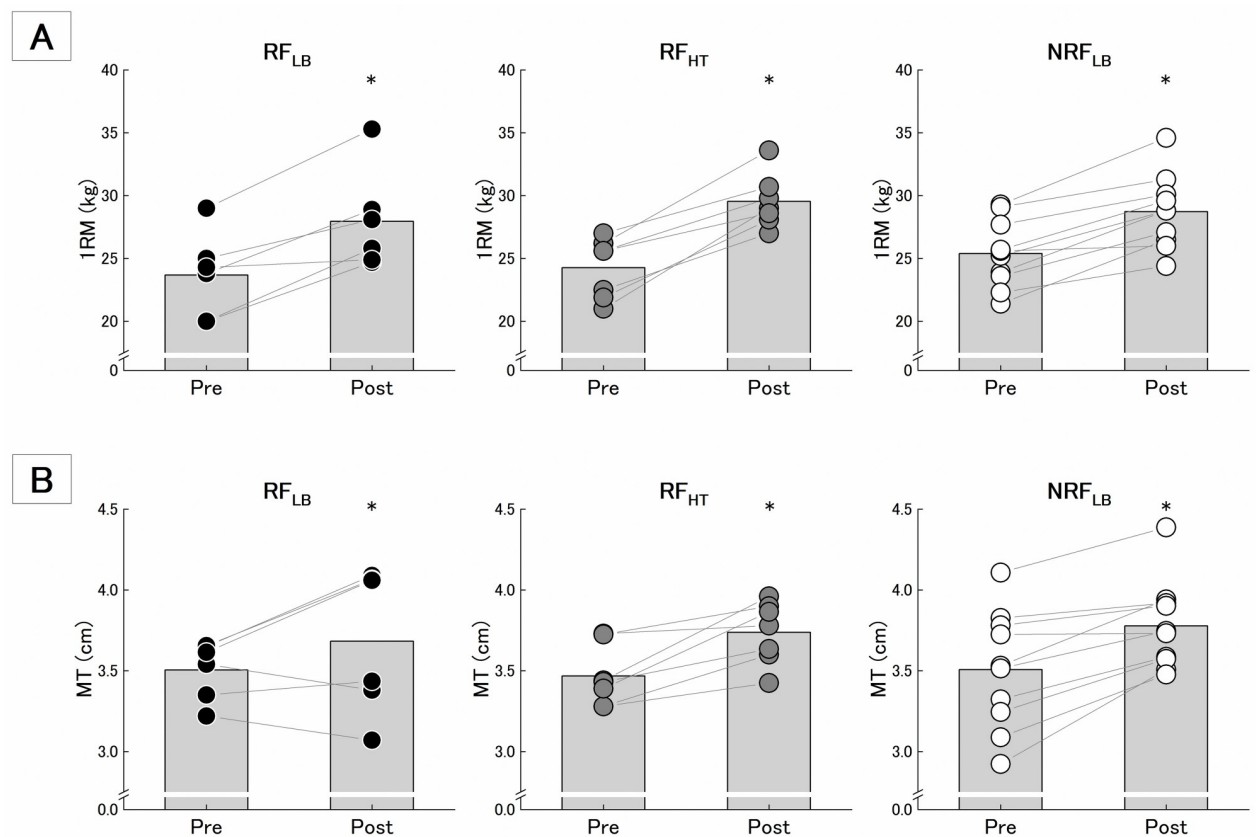

**Fig 3.** Mean and individual changes in 1RM (A) and MT (B). The black-filled bar graph and the filled circle plots were presented as means and individual's values. * indicates the difference between time is significant at p < 0.05.

magnitude of effect size for changes in the variables having significant effect of time varied among the three training groups. It was doubtful that there is any power of detection because a total sample size of 42 participants was required according to power analysis. Due to the spread of COVID-19, however, additional subjects could not be recruited. So, the statistical power (1-β) was computed with total sample size (N = 23), effect size calculated from partial $\eta^2$ and the significant level (p<0.05) for 1RM, muscle size, MVC, $F_0/CSA_{index}$ and $P_{max}/CSA_{index}$. As the results, the statistical power was more than 0.8 in all variables, indicating that the effects of the training interventions adopted in the present study had a large effect size. Therefore, the sample size may be sufficient to achieve the required statistical power. In addition, 1RM, MVC, and $CSA_{index}$ tended to increase, but $F_0/CSA_{index}$ and $P_{max}/CSA_{index}$ showed opposite tendencies. Training history and current level of muscular fitness directly influences on the content of training variables designed and the potential magnitude of training adaptations [32]. When designing RT programs for athletes, therefore, strength and conditioning coaches are always required to manipulate the training variables for maximizing the muscle function of the practitioners beyond their current levels. Thus, the observed differences between the groups in the magnitude of effect size and discrepancy between the training-induced changes in the parameters of muscle function are meaningful to discuss the differences between the programs adopted here in the effects on the muscle size and function in athletes, being required to develop high muscular strength and power, such as judo athletes.

**Table 1. Descriptive data on 1RM, MT and CSA$_{index}$.**

| Variables | Groups | Pre | Post | %Δ | g | CI | two-way ANOVA[a] | | |
|---|---|---|---|---|---|---|---|---|---|
| | | | | | | | time | group | Interaction |
| 1RM (kg) | RF$_{LB}$ | 23.7 ± 3.4 | 28.0 ± 4.0 | 18.4 | 1.06 | -0.09 ~ 2.39 | | | |
| | RF$_{HT}$ | 24.3 ± 2.4 | 29.5 ± 2.1 | 22.4 | 2.18 | 0.93 ~ 3.73 | <0.001 | 0.68 | 0.12 |
| | NRF$_{LB}$ | 25.4 ± 2.7 | 28.7 ± 2.9 | 13.4 | 1.13 | 0.23 ~ 2.14 | | | |
| MT (cm) | RF$_{LB}$ | 3.5 ± 0.2 | 3.7 ± 0.4 | 4.8 | 0.49 | -0.63 ~ 1.68 | | | |
| | RF$_{HT}$ | 3.5 ± 0.2 | 3.7 ± 0.2 | 7.9 | 1.33 | 0.23 ~ 2.61 | <0.001 | 0.93 | 0.66 |
| | NRF$_{LB}$ | 3.5 ± 0.4 | 3.8 ± 0.3 | 8.1 | 0.80 | -0.08 ~ 1.75 | | | |
| CSA$_{index}$ (cm$^2$) | RF$_{LB}$ | 9.7 ± 1.0 | 10.8 ± 2.5 | 10.4 | 0.54 | -0.58 ~ 1.74 | | | |
| | RF$_{HT}$ | 9.5 ± 1.0 | 11.0 ± 1.1 | 16.7 | 1.33 | 0.22 ~ 2.61 | <0.001 | 0.92 | 0.78 |
| | NRF$_{LB}$ | 9.8 ± 2.0 | 11.3 ± 1.7 | 17.2 | 0.78 | -0.10 ~ 1.73 | | | |

Pre and post values are mean ± SD.

1RM, one repetition maximum; MT, muscle thickness; CSA$_{index}$, CSA estimated from MT;

%Δ, the average value of relative change; g, Hedges's g value; CI, 95% confidence intervals;

[a], p value of two-way ANOVA.

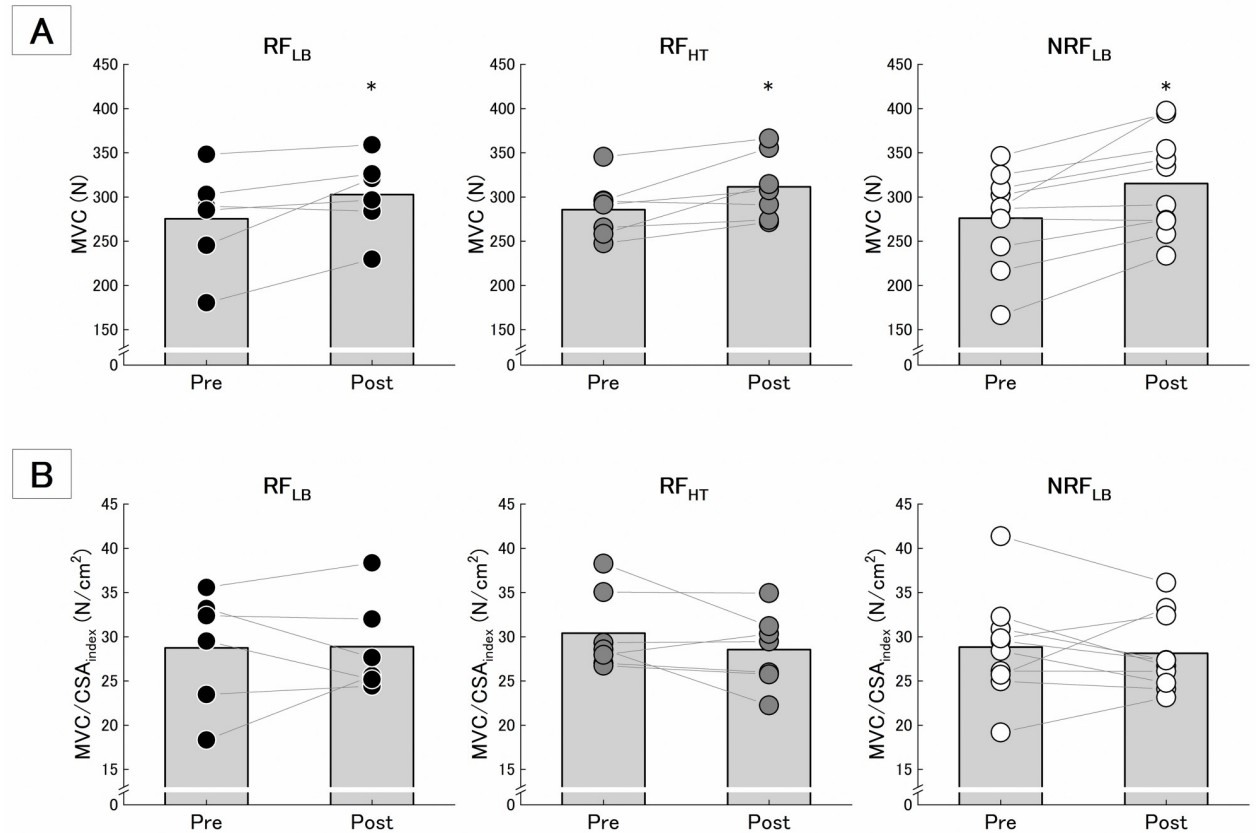

**Fig 4.** Mean and individual changes in MVC (A) and MVC/CSAindex (B). The black-filled bar graph and the filled circle plots were presented as means and individual's values. * indicates the difference between time is significant at p < 0.05.

**Table 2. Descriptive data on MVC, MVC/CSA$_{index}$ and RFD$_{max}$.**

| Variables | Groups | Pre | Post | %Δ | g | CI | two-way ANOVA[a] | | |
|---|---|---|---|---|---|---|---|---|---|
| | | | | | | | time | group | Interaction |
| MVC (N) | RF$_{LB}$ | 275.4 ± 57.1 | 302.7 ± 44.2 | 11.8 | 0.49 | -0.62 ~ 1.69 | | | |
| | RF$_{HT}$ | 285.6 ± 32.7 | 311.6 ± 37.3 | 9.3 | 0.69 | -0.35 ~ 1.83 | <0.001 | 0.93 | 0.60 |
| | NRF$_{LB}$ | 276.2 ± 53.8 | 315.3 ± 57.4 | 15.4 | 0.67 | -0.20 ~ 1.61 | | | |
| MVC/CSA$_{index}$ (N/cm$^2$) | RF$_{LB}$ | 28.7 ± 6.6 | 28.9 ± 5.4 | 3.2 | 0.02 | -1.11 ~ 1.15 | | | |
| | RF$_{HT}$ | 30.4 ± 4.5 | 28.6 ± 4.2 | -5.6 | -0.40 | -1.49 ~ 0.63 | 0.38 | 0.91 | 0.69 |
| | NRF$_{LB}$ | 28.8 ± 5.8 | 28.1 ± 4.3 | -0.6 | -0.13 | -1.01 ~ 0.74 | | | |
| RFD$_{max}$ (N/s) | RF$_{LB}$ | 3184.7 ± 679.4 | 2815.6 ± 494.0 | -8.8 | -0.57 | -1.79 ~ 0.54 | | | |
| | RF$_{HT}$ | 3598.8 ± 698.5 | 3676.1 ± 762.5 | 3.2 | 0.10 | -0.94 ~ 1.15 | 0.72 | 0.17 | 0.09 |
| | NRF$_{LB}$ | 3257.2 ± 565.2 | 3704.9 ± 851.7 | 15.5 | 0.59 | -0.28 ~ 1.52 | | | |

Pre and post values are mean ± SD.

MVC, maximal voluntary contraction; CSA$_{index}$, CSA estimated from MT; RFD, rate of force development;

%Δ, the average value of relative change; g, Hedges's g value; CI, 95% confidence intervals;

[a], p value of two-way ANOVA.

**Table 3. Descriptive data on the parameters derived from F-V relation.**

| Variables | Groups | Pre | Post | %Δ | g | CI | two-way ANOVA[a] | | |
|---|---|---|---|---|---|---|---|---|---|
| | | | | | | | time | group | Interaction |
| F$_0$ (N) | RF$_{LB}$ | 306.1 ± 41.5 | 307.2 ± 31.4 | 3.2 | 0.03 | -1.10 ~ 1.16 | | | |
| | RF$_{HT}$ | 327.4 ± 53.3 | 325.6 ± 47.1 | 0.9 | -0.03 | -1.08 ~ 1.01 | 0.56 | 0.28 | 0.63 |
| | NRF$_{LB}$ | 285.8 ± 52.2 | 305.9 ± 41.6 | 8.4 | 0.41 | -0.46 ~ 1.31 | | | |
| V$_0$ (ms) | RF$_{LB}$ | 2.2 ± 0.5 | 2.2 ± 0.3 | 6.7 | 0.09 | -1.03 ~ 1.23 | | | |
| | RF$_{HT}$ | 2.2 ± 0.4 | 2.2 ± 0.4 | 1.6 | 0.07 | -0.97 ~ 1.12 | 0.98 | 0.21 | 0.80 |
| | NRF$_{LB}$ | 2.0 ± 0.3 | 1.9 ± 0.4 | -3.0 | -0.22 | -1.11 ~ 0.65 | | | |
| P$_{max}$ (W) | RF$_{LB}$ | 164.9 ± 29.3 | 170.3 ± 16.3 | 5.4 | 0.21 | -0.91 ~ 1.36 | | | |
| | RF$_{HT}$ | 176.8 ± 36.0 | 177.5 ± 33.7 | 2.2 | 0.02 | -1.03 ~ 1.07 | 0.49 | 0.15 | 0.92 |
| | NRF$_{LB}$ | 143.6 ± 34.2 | 149.1 ± 42.9 | 4.1 | 0.14 | -0.74 ~ 1.02 | | | |
| F-V$_{slope}$ (N/ms) | RF$_{LB}$ | 148.5 ± 50.4 | 140.2 ± 29.8 | 8.1 | -0.19 | -1.34 ~ 0.93 | | | |
| | RF$_{HT}$ | 154.6 ± 35.2 | 152.7 ± 36.7 | 0.3 | -0.05 | -1.10 ~ 1.00 | 0.83 | 0.77 | 0.55 |
| | NRF$_{LB}$ | 144.6 ± 32.7 | 161.6 ± 30.1 | 15.8 | 0.52 | -0.35 ~ 1.44 | | | |
| F$_0$/CSA$_{index}$ (N/cm$^2$) | RF$_{LB}$ | 32.1 ± 6.1 | 29.8 ± 7.3 | -5.6 | -0.31 | -1.48 ~ 0.80 | | | |
| | RF$_{HT}$ | 35.1 ± 7.5 | 29.7 ± 3.8 | -12.9 | -0.85 | -2.01 ~ 0.20 | 0.01 | 0.24 | 0.53 |
| | NRF$_{LB}$ | 29.8 ± 4.9 | 27.3 ± 2.5 | -6.7 | -0.62 | -1.55 ~ 0.26 | | | |
| P$_{max}$/CSA$_{index}$ (W/cm$^2$) | RF$_{LB}$ | 17.3 ± 4.0 | 16.4 ± 3.6 | -2.6 | -0.20 | -1.36 ~ 0.91 | | | |
| | RF$_{HT}$ | 18.7 ± 3.4 | 16.1 ± 2.0 | -11.9 | -0.89 | -2.06 ~ 0.17 | 0.01 | 0.02 | 0.55 |
| | NRF$_{LB}$ | 14.8 ± 2.1 | 13.1 ± 2.2 | -10.6 | -0.75 | -1.69 ~ 0.13 | | | |

Pre and post values are mean ± SD.

F$_0$, theoretical maximum force; V$_0$, theoretical maximum velocity; P$_{max}$, theoretical maximum power;

F-V$_{slope}$, slope were calculated from force-velocity relation; CSA$_{index}$, CSA estimated from MT;

%Δ, the average value of relative change; g, Hedges's g value; CI, 95% confidence intervals;

[a], p value of two-way ANOVA.

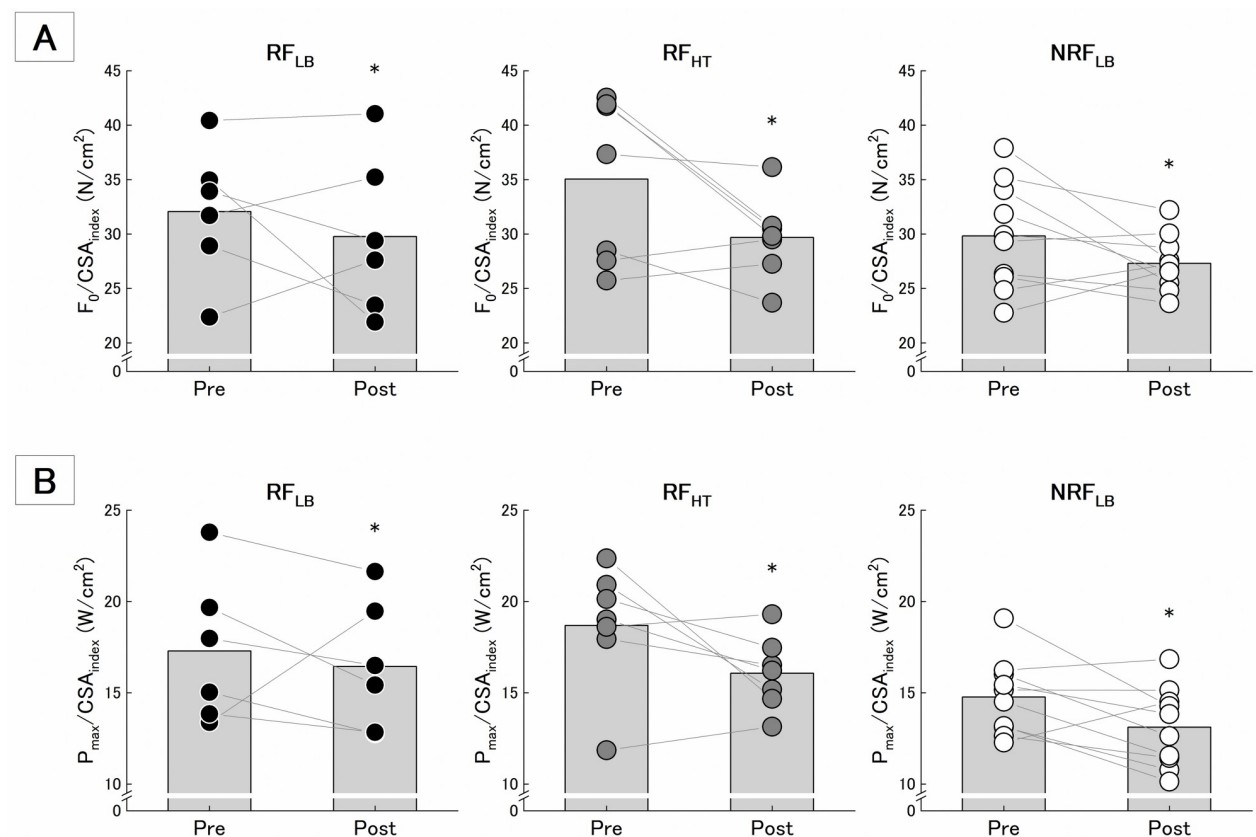

**Fig 5.** Mean and individual changes in F0/CSAindex (A) and Pmax/CSAindex (B). The black-filled bar graph and the filled circle plots were presented as means and individual's values. * indicates the difference between time is significant at p < 0.05.

Magnitude of effect sizes for the changes in MVC and MT for $RF_{LB}$ was smaller than those for $RF_{HT}$ and $NRF_{LB}$ (Table 2). This might be due to the influence of dose response relationship on either muscle strength or size. Peterson et al. [32, 33] have reported that for athlete populations, maximal strength gains are elicited at a training intensity of 85% of 1RM, 2 days per week, and with a training volume of 8 sets per muscle group. In their findings, strength gain tended to decrease when the training volume exceeds 8 sets per muscle group. In addition, a systematic review concerning the influence of weekly sets per muscle group on muscle size has shown that compared to "low" (<12 weekly sets) and "high" (>20 weekly sets) volume, "moderate" volume (12–20 weekly sets) is an optimum standard recommendation for increasing muscle hypertrophy in young trained men [34]. In the present study, the training volume of $RF_{LB}$ was greatest among the three groups. As a result of the combination of the prescribed training program and judo-specific training, therefore, it seems that for $RF_{LB}$, the total training volume might have become a factor resulting in small gains in either MVC or MT in this group.

While 1RM, MVC, and MT had significant effect of time, all parameters derived from F-V relation did not. This indicates that at least for the current participants, the training-induced gain in force generation capacity did not link to that in power generation capacity. There is a fundamental relationship between muscle strength and power generation capacity [35]. Thus, individuals who possess a high level of muscular strength can develop greater muscular power [36]. Joint torque in humans is closely related to muscle volume [37]. As a training strategy for

long-term development of muscular power, therefore, enhancing and maintaining muscularity and force generation capacity become basic approach [35]. However, the current results deny the expectation derived from the fundamental relationship between muscle strength and power generation capacity. From the findings of Mangine et al. [38] who compared between the effects of high-intensity (90% of 1RM) and high-volume (70% of 1RM) RTs in resistance-trained men, high-intensity RT was more advantages than high-volume RT to improve isometric MVC and RFD. In their results, however, changes in barbell velocity during 1RM assessments were similar between the two protocols, suggesting that the superiority in improving isometric MVC and RFD does not translate toward greater benefits for providing greater gains in power output at a given load. This finding indirectly supports our results.

It is known that strength gain in the early stage of training period is mainly associated with neural adaptation such as improved co-ordination or learning [39] and increased motor unit activation [40]. In this phase, the magnitude of muscle hypertrophic change is smaller than that of MVC and consequently the ratio of MVC to muscle CSA increases [41, 42]. As described earlier, it was expected that high-load RT leading to repetition failure could be a potential modality yielding high activation of motor unit pool of the exercising muscles, contributing to enhance the ratio of MVC to muscle size. In the current results, however, MVC/$CSA_{index}$ had no significant effect of time, suggesting that the gains in MVC would be mainly attributable to those in muscle size. The lack of significant change in MVC/$CSA_{index}$ might be due to the high neuromuscular function in judo athletes. In our previous study, judo athletes have a greater MVC/$CSA_{index}$, compared to gymnasts [8]. Furthermore, well-trained men can activate the motor cortex more during voluntary muscle contraction, compared to untrained men [43]. A previous study has demonstrated that as a result of a 3-week isometric training to volitional failure, the individuals with high MVC/$CSA_{index}$ have a smaller strength gain compared to those with low MVC/$CSA_{index}$ [44]. In addition, the training-induced change in muscle size is negatively associated with that in MVC/$CSA_{index}$, suggesting that as muscle adaptation to resistance training, the practitioners with less improvement in MVC/$CSA_{index}$ show a large gain in muscle size [45]. Considering these findings, it is assumed that judo athletes might have less room for improvement in MVC/$CSA_{index}$ through the resistance training and consequently they might have shown a large hypertrophic change.

The $F_0$/$CSA_{index}$ and $P_{max}$/$CSA_{index}$ had significant effects of time with tendencies of decrease. The magnitude of the negative changes was large in $RF_{HT}$ and moderate in $NRF_{LB}$ (Table 3). The discrepancy between the RT-induced gains of muscle size and dynamic strength have been already reported by some previous studies in which muscle strength has been determined in different action and contraction manners from those adopted as training protocols. For example, Kawakami et al. [46], who adopted 16-week unilateral RT at 80% of 1RM for elbow extensor muscles, has observed significant decrease in specific tension in a range of preset velocities of -200 to 0 deg/s, determined as the isometric and dynamic eccentric forces at the tendon divided by physiological CSA. In addition, Tanimoto and Ishii [47], who examined the effects of low-intensity RT with slow movement and tonic force generation, reported that RT at ~50% of 1RM produced significant gains in quadriceps femoris CSA and isometric torque and isokinetic torque at preset angular velocity of 90 deg/s. In their results, however, no significant gains were found in isokinetic torques at preset velocities of 200 and 300 deg/s. Furthermore, some cross-sectional studies have provided evidence suggesting that strength trained individuals with high muscularity show lower dynamic strength relative to muscle size compared to untrained control [48, 49]. In addition, the ratios of dynamic strength of elbow flexors, determined at preset angular velocities of ~ 300 deg/s, to elbow flexor CSA have been shown to negatively correlated to the CSA in populations including strength trained individuals [48, 49]. The influences of changes in muscle geometry associated with muscle hypertrophy,

inducing smaller net force acting tendon tissues [46, 49, 50] and low neuromuscular activities during the maximal dynamic tasks [8] have been considered as reasons for the negative association of muscle size to dynamic strength relative to muscle size.

It is unknown whether the factors described above can be applied to the current results on the ratios of $F_0$ and $P_{max}$ to $CSA_{index}$. Nakatani et al. [8] observed that the activity levels of the biceps brachii during maximal elbow flexions were higher in judo athletes than in gymnasts. Judo athletes as well as gymnasts are characterized by a predominant muscular development in the upper limb [25]. From the results of Nakatani et al. [8], the ratios of $F_0$ and $P_{max}$ to the $CSA_{index}$ of elbow flexors were also higher in the judo athletes than in gymnasts, while the $CSA_{index}$ of elbow flexors was similar between the two athletic groups. Thus, there is a possibility that for the judo athletes examined here, the room of neuromuscular adaptation to training programs adopted here might have been small, and consequently induced medium to large decreases in $F_0/CSA_{index}$ and $P_{max}/CSA_{index}$ in $RF_{HT}$ and $NRF_{LB}$ which showed hypertrophic changes. In any case, the current results as well as the previous findings cited here suggest that for strength trained athletes with high muscularity, further gains in muscle size are not always positively affect for improving explosive force and power generation capacity, expressed as the values relative to muscle size.

We should comment some limitations, relating to the experimental design adopted here. The present study recruited judo athletes as the participants, having greater musculature and strength and power generation capacity compared to other athletic populations [8, 25]. Thus, there is a possibility that for the judo athletes, the intervention period (6 weeks) might have been too short to produce gains in power generation capacity. However, previous studies, which examined the impact of strength level on adaptations to a 10-week ballistic and power training, have observed significant gains in power output during unloaded jump squat after 5 weeks even in stronger group [36, 51]. In addition, some studies have observed significant gains in sprint and jump performance and the measured power variables in athletes as results of resistance training programs for six weeks [52–55]. These suggest that the intervention period set in this study would not be a decisive reason for the lack of significant effect of time in the parameters related to force-velocity relation. $RF_{HT}$ performed the exercise task with a constant tempo rather than the maximum intended velocity to protect the participants from the risk of injury. The effects of heavy load training with ballistic contractions on muscle strength and power are subjects in further investigation. Furthermore, a high inter-subject variability exits regarding the maximum number of repetitions performed with and specific relative load, therefore 20 repetitions might have entailed different level of effort between participants of $NRF_{LB}$. Second, it should be considered that in the present study, all participants conducted the prescribed training program as an adjunct exercise task to judo-specific training. For athletic populations, many studies have observed significant gains in muscular power output and/or explosive motor performances by adopting tapering phase in which training volume, intensity, and or frequency are progressively reduced [56–60]. Thus, we cannot rule out a possibility that the power generation capacity of the current participants might have been improved if an experimental design including tapering phase is adopted. Further study is needed to clarify this. Third, there are various training variables that contribute to increase muscle mass and strength, and it is possible that the effects of different conditions have either masked each other or created synergistic effects. Recent meta-analyses have concluded that for strength improvements, increases in strength are superior in high-load RT programs [22] and that multi-set protocols are more effective than single-set ones [61]. Training fast produces greater strength increases than training slow; however, there does not appear to be any additional benefit of training with both multiple sets and fast contractions [62]. Resistance exercise performed to failure elevates muscle protein synthesis regardless of training

volume (sets × reps) or % one repetition maximum load [63, 64]. Furthermore, Sampson and Groeller [65] conducted a study where they varied the speed and number of repetitions in elbow flexion exercises, and reported that, despite differences in training volume, no significant differences were found in the effects on muscle mass and strength. They suggest that repetition failure and training volume may be of less importance for the development of muscle hypertrophy and strength when the characteristics of the muscle activation are manipulated. In this study, despite the fact that training volume differed among the three groups, both muscle strength and muscle hypertrophy increased similarly regardless of any of the groups. Similar skeletal muscle adaptations can be gained with rapid muscle activation in the absence of repetition failure and a concurrent reduction in the total exercise volume. Hence, we cannot rule out a possibility that the current results may differ if the training volumes are aligned. Fourth, the study had a relatively small sample size and thus may have been somewhat underpowered to detect significant changes between groups in certain outcomes.

In conclusion, the findings obtained in this study indicate that ballistic light-load and traditional heavy-load resistance training programs, leading to non-repetition failure and repetition failure, respectively, can be modalities for improving muscle size and isometric strength in judo athletes, but these do not improve power generation capacity.

## Practical applications

The findings obtained here did not support our hypothesis that traditional heavy-load RT leading to repetition failure could be a potential modality to maximize power generation capacity in strength-trained athletes such as judo athletes. However, we can say that the traditional heavy-load RT leading to repetition failure would be an efficient modality for athletes specializing sport events in which great muscle mass and strength are essential to achieve high performance. In the present study, the average value of the total training volume for $RF_{HT}$ was lowest among the three groups set in the present study, and the corresponding value was 34.1% and 52.2% for those for $RF_{LB}$ and $NRF_{LB}$, respectively. In spite of the differences in the total training volume, MT and MVC had significant main effect of time without interaction of time and groups. Taking this into account together with the fact that the present study was conducted during the in-season phase of judo, it is likely that for strength-trained athletes such as judo athletes, traditional heavy-load RT leading to repetition failure can be an efficient modality to increase muscle size and strength with low training volume.

## Supporting information

**S1 File.**
(XLSX)

**S2 File.**
(PDF)

## Acknowledgments

The authors wish to express their gratitude to the students of the National Institute of Fitness and Sports in Kanoya for their contribution to this study.

## Author Contributions

**Conceptualization:** Miyuki Nakatani, Yohei Takai, Hiroaki Kanehisa.

**Data curation:** Miyuki Nakatani, Hiroaki Kanehisa.

**Formal analysis:** Miyuki Nakatani, Yohei Takai, Hiroaki Kanehisa.

**Investigation:** Miyuki Nakatani, Yohei Takai, Hiroaki Kanehisa.

**Methodology:** Miyuki Nakatani, Yohei Takai, Hiroaki Kanehisa.

**Project administration:** Miyuki Nakatani, Yohei Takai, Hiroaki Kanehisa.

**Resources:** Yohei Takai.

**Software:** Yohei Takai.

**Supervision:** Miyuki Nakatani, Yohei Takai, Hiroaki Kanehisa.

**Validation:** Miyuki Nakatani.

**Visualization:** Miyuki Nakatani.

**Writing – original draft:** Miyuki Nakatani.

**Writing – review & editing:** Miyuki Nakatani, Yohei Takai, Hiroaki Kanehisa.

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
