## [Decision Letter · Decision Letter 0]

9 Apr 2024

PONE-D-23-40875Resistance training leading to repetition failure increases muscle strength and size, but not power-generation capacity in judo athletes.PLOS ONE

Dear Dr. Nakatani,

Thank you for submitting your manuscript to PLOS ONE. After careful consideration, we feel that it has merit but does not fully meet PLOS ONE’s publication criteria as it currently stands. Therefore, we invite you to submit a revised version of the manuscript that addresses the points raised during the review process.

**ACADEMIC EDITOR: **Dear Authors,one expert in the field reviewed your manuscript retrieving several issues you should address during the revision process. Please submit your revised manuscript by May 24 2024 11:59PM. If you will need more time than this to complete your revisions, please reply to this message or contact the journal office at plosone@plos.org. Please include the following items when submitting your revised manuscript:A rebuttal letter that responds to each point raised by the academic editor and reviewer(s). You should upload this letter as a separate file labeled 'Response to Reviewers'.A marked-up copy of your manuscript that highlights changes made to the original version. You should upload this as a separate file labeled 'Revised Manuscript with Track Changes'.An unmarked version of your revised paper without tracked changes. You should upload this as a separate file labeled 'Manuscript'.

We look forward to receiving your revised manuscript.

Kind regards,

Emiliano Cè

Academic Editor

PLOS ONE

Journal Requirements:

Reviewers' comments:

Reviewer's Responses to Questions

**Comments to the Author**

1. Is the manuscript technically sound, and do the data support the conclusions?

Reviewer #1: Partly

2. Has the statistical analysis been performed appropriately and rigorously? 

Reviewer #1: No

3. Have the authors made all data underlying the findings in their manuscript fully available?

Reviewer #1: Yes

4. Is the manuscript presented in an intelligible fashion and written in standard English?

Reviewer #1: Yes

5. Review Comments to the Author

Reviewer #1: In this study neuromuscular adaptations to three 6-week elbow flexor training programs differing in intensity of load and muscular failure (30% 1RM leading to failure, 30% 1RM not lading to failure, and 80% 1RM performed to failure) were contrasted in a small sample of judo athletes. I have the following concerns about this manuscript:

1. Sample size of each group is very small (between 6 and 10)

2. The authis must provide a detailed description of all the procedures.

3. Statistical analysis should be improved (see specific comments)

4. Discussion section is too long, being similar to a narrative review. In my opinion it should be rewritten to be focused on the main findings of the results.

SPECIFIC COMMENTS (L= Line)

L50. I´d suggest using the term heavy-load and light-load instead heavier/lighter-load.

L61-65. This statement is not true. There are several studies that show how RT not leading to muscular failure optimizes power gains.

L85-86. The authors should reword this sentence.

L102-107.The wording of this paragraph is confusing to me. I would suggest reviewing it. On the other hand, what do you mean by ballistic? Throwing the load? At maximum voluntary velocity? Wasn´t maximal intended velocity performed with 80% 1RM load?

L 110. This period corresponds to COVID-19 pandemic breakdown, doesn´t it? Were the participants following their usual training program? On the other hand, the sample size is quite limited and should be commented a s a limitation.

L112-116. Please specify the description of the sample: did they perform resistance training? In this case, what was training frequency? Did they include the tested exercise in their training program?

L147. Please, replace are by were.

L 147150. It is strange that the athletes were able to maintain this pace until muscular failure. It is well known that velocity decreases with fatigue, so it is very strange to perform repetitions until muscular failure without losing velocity. Why RFHT was not performed to the maximum intended velocity? How could this have affected your results?

See for example: https://www.ncbi.nlm.nih.gov/pmc/articles/PMC9613575/;
https://pubmed.ncbi.nlm.nih.gov/8444715/

L152-153. A high inter-subject variability exits regarding the maximum number of repetitions performed with and specific relative load, therefore 20 repetitions might have entailed different level of effort between participants. Did you test maximum number of repetitions with this load? If this was not the case, you should discuss this point as a limitation of your study.

L163-165. I do not understand this point. This a surrogate estimation of mechanical impulse, isn´t it? On the other hand, RT with light loads was performed at maximum intended velocity, therefore time under tension varied throughout each set.

L249. How did you obtain angular velocity?

STATISTICAL ANALYSIS:

L265. You had three groups and therefore one-way ANOVA should also be used for baseline measurements?

L270-271. Sample size is really small, and therefore the use of non-parametric statistic should be recommended. Did you test normality and homocedasticity of your variables?

L 275. Please, report 95% confidence intervals for Hedge´s g.

L282-285. This result is trivial given the experimental design. On the other hand it highlights that differences or lack of differences may be caused by differences in volume instead of differences in loads. In my opinion this is the main limitation of this study since the independent variable is not well controlled.

L290-292. Please note that figure 3A sems to show simple contrasts that are not supported on significant interactions. Pairwise comparisons that are not based on significant interaction should be removed throughout the manuscript. In this regard, discussion and conclusions must be properly adapted.

L312-314. Was there a tendency or a significant main effect of time? Please, clarify this point.

Table 3. Units of slope should be reported

DISCUSSION

Discussion is too long considering your findings: statistically speaking, the lack of interaction suggest that training conditions induced similar adaptations, despite the differences in volume. In my opinion the discussion must be rewritten and focused on this point. Finally, the interpretation of your results from the perspective of differences in volume should not be limited to conclusions.

6. PLOS authors have the option to publish the peer review history of their article (what does this mean?). If published, this will include your full peer review and any attached files.

Reviewer #1: No

---

## [Author Response · Author response to Decision Letter 0]

20 May 2024

Dear Emiliano Cè

Academic Editor

PLOS ONE

Very thanks for your consideration and valuable suggestions to improve our manuscript. We read the reviewer’s comments very carefully and made the following changes in accordance with the comments and suggestions of the reviewer.

We hope that the revised manuscript answers the questions and comments of the reviewers.

---

## [Decision Letter · Decision Letter 1]

10 Jun 2024

PONE-D-23-40875R1Resistance training leading to repetition failure increases muscle strength and size, but not power-generation capacity in judo athletes.PLOS ONE

Dear Dr. Nakatani,

Thank you for submitting your manuscript to PLOS ONE. After careful consideration, we feel that it has merit but does not fully meet PLOS ONE’s publication criteria as it currently stands. Therefore, we invite you to submit a revised version of the manuscript that addresses the points raised during the review process.

**ACADEMIC EDITOR: **Dear Authors,one expert in the field reviewed your manuscript detecting some minor issues you should consider while revising your study. Please submit your revised manuscript by Jul 25 2024 11:59PM. If you will need more time than this to complete your revisions, please reply to this message or contact the journal office at plosone@plos.org. Please include the following items when submitting your revised manuscript:A rebuttal letter that responds to each point raised by the academic editor and reviewer(s). You should upload this letter as a separate file labeled 'Response to Reviewers'.A marked-up copy of your manuscript that highlights changes made to the original version. You should upload this as a separate file labeled 'Revised Manuscript with Track Changes'.An unmarked version of your revised paper without tracked changes. You should upload this as a separate file labeled 'Manuscript'.If applicable, we recommend that you deposit your laboratory protocols in protocols.io to enhance the reproducibility of your results. Protocols.io assigns your protocol its own identifier (DOI) so that it can be cited independently in the future. For instructions see: https://journals.plos.org/plosone/s/submission-guidelines#loc-laboratory-protocols. Additionally, PLOS ONE offers an option for publishing peer-reviewed Lab Protocol articles, which describe protocols hosted on protocols.io. Read more information on sharing protocols at https://plos.org/protocols?utm_medium=editorial-email&utm_source=authorletters&utm_campaign=protocols.

We look forward to receiving your revised manuscript.

Kind regards,

Emiliano Cè

Academic Editor

PLOS ONE

Journal Requirements:

Reviewers' comments:

Reviewer's Responses to Questions

**Comments to the Author**

1. If the authors have adequately addressed your comments raised in a previous round of review and you feel that this manuscript is now acceptable for publication, you may indicate that here to bypass the “Comments to the Author” section, enter your conflict of interest statement in the “Confidential to Editor” section, and submit your "Accept" recommendation.

Reviewer #1: (No Response)

2. Is the manuscript technically sound, and do the data support the conclusions?

Reviewer #1: Partly

3. Has the statistical analysis been performed appropriately and rigorously? 

Reviewer #1: Yes

4. Have the authors made all data underlying the findings in their manuscript fully available?

Reviewer #1: Yes

5. Is the manuscript presented in an intelligible fashion and written in standard English?

Reviewer #1: Yes

6. Review Comments to the Author

Reviewer #1: The authors have correctly addressed most of my previous comments. Nevertheless, there are still a few aspects that should be improved:

Lines 64-65: “A few studies have examined the effect of resistance training leading to repetition failure on power generating capacity in athletes.” Please provide some references to support this statement.

Line 106: Please replace "was" with "is." On the other hand, was the exercise with heavy load performed at submaximal velocity? Please clarify this. Intended velocity is a key point for specific adaptations, and thus, the authors should consider this aspect in the discussion section.

Line 342 (Power analysis): This is an "observed" power analysis. On the other hand, the specific test for which you performed this analysis is not reported. It seems that it was calculated for a repeated measures design, not for a between-within interaction. How many participants in each group would be needed to achieve a statistical power of 0.8 for detecting a medium effect size in the interaction? Provide a detailed description of this power calculation and report the software that you used.

7. PLOS authors have the option to publish the peer review history of their article (what does this mean?). If published, this will include your full peer review and any attached files.

Reviewer #1: No

---

## [Author Response · Author response to Decision Letter 1]

24 Jun 2024

Thank you for your valuable suggestions to improve our manuscript. We carefully reviewed the reviewer’s comments and made the necessary revisions. The changes are highlighted in the manuscript with page and line numbers for each comment. We hope the revised manuscript addresses all the reviewer's concerns.

---

## [Decision Letter · Decision Letter 2]

11 Jul 2024

Resistance training leading to repetition failure increases muscle strength and size, but not power-generation capacity in judo athletes.

PONE-D-23-40875R2

Dear Dr. Nakatani,

We’re pleased to inform you that your manuscript has been judged scientifically suitable for publication and will be formally accepted for publication once it meets all outstanding technical requirements.

Kind regards,

Emiliano Cè

Academic Editor

PLOS ONE

Additional Editor Comments (optional):

Reviewers' comments:

Reviewer's Responses to Questions

**Comments to the Author**

1. If the authors have adequately addressed your comments raised in a previous round of review and you feel that this manuscript is now acceptable for publication, you may indicate that here to bypass the “Comments to the Author” section, enter your conflict of interest statement in the “Confidential to Editor” section, and submit your "Accept" recommendation.

Reviewer #1: All comments have been addressed

2. Is the manuscript technically sound, and do the data support the conclusions?

Reviewer #1: Yes

3. Has the statistical analysis been performed appropriately and rigorously? 

Reviewer #1: Yes

4. Have the authors made all data underlying the findings in their manuscript fully available?

Reviewer #1: Yes

5. Is the manuscript presented in an intelligible fashion and written in standard English?

Reviewer #1: Yes

6. Review Comments to the Author

Reviewer #1: (No Response)

7. PLOS authors have the option to publish the peer review history of their article (what does this mean?). If published, this will include your full peer review and any attached files.

Reviewer #1: No

---

## [Editor Report · Acceptance letter]

17 Jul 2024

PONE-D-23-40875R2 

PLOS ONE

Dear Dr. Nakatani, 

I'm pleased to inform you that your manuscript has been deemed suitable for publication in PLOS ONE. Congratulations! Your manuscript is now being handed over to our production team.

Kind regards, 

on behalf of

Prof. Emiliano Cè 

Academic Editor

PLOS ONE